# Synthesis and Exon-Skipping Properties of a 3′-Ursodeoxycholic Acid-Conjugated Oligonucleotide Targeting *DMD* Pre-mRNA: Pre-Synthetic versus Post-Synthetic Approach

**DOI:** 10.3390/molecules26247662

**Published:** 2021-12-17

**Authors:** Elena Marchesi, Matteo Bovolenta, Lorenzo Preti, Massimo L. Capobianco, Kamel Mamchaoui, Monica Bertoldo, Daniela Perrone

**Affiliations:** 1Department of Environmental and Prevention Sciences, University of Ferrara, 44121 Ferrara, Italy; mrclne@unife.it; 2Department of Translational Medicine, University of Ferrara, 44121 Ferrara, Italy; bvlmtt@unife.it; 3Department of Chemical, Pharmaceutical and Agricultural Sciences, University of Ferrara, 44121 Ferrara, Italy; prtlnz@unife.it (L.P.); brtmnc@unife.it (M.B.); 4Institute of Organic Synthesis and Photoreactivity, Italian National Research Council, 40129 Bologna, Italy; massimo.capobianco@isof.cnr.it; 5Centre de Recherche en Myologie, Institut de Myologie, Sorbonne Université, Inserm, F-75013 Paris, France; kamel.mamchaoui@upmc.fr

**Keywords:** antisense oligonucleotide, conjugation, bile acid, ursodeoxycholic acid, solid support, solid phase synthesis, exon-skipping, dystrophin

## Abstract

Steric blocking antisense oligonucleotides (ASO) are promising tools for splice modulation such as exon-skipping, although their therapeutic effect may be compromised by insufficient delivery. To address this issue, we investigated the synthesis of a 20-mer 2′-OMe PS oligonucleotide conjugated at 3′-end with ursodeoxycholic acid (UDCA) involved in the targeting of human *DMD* exon 51, by exploiting both a pre-synthetic and a solution phase approach. The two approaches have been compared. Both strategies successfully provided the desired ASO 51 3′-UDC in good yield and purity. It should be pointed out that the pre-synthetic approach insured better yields and proved to be more cost-effective. The exon skipping efficiency of the conjugated oligonucleotide was evaluated in myogenic cell lines and compared to that of unconjugated one: a better performance was determined for ASO 51 3′-UDC with an average 9.5-fold increase with respect to ASO 51.

## 1. Introduction

Antisense oligonucleotides (ASO) are therapeutically attractive short nucleic acid chains exploited for gene modulation by targeting specific sequences in pre-mRNA or mRNA [1]. Their mode of action depends on the nature of target sequence and the chemistry used [2]. Several ASO chemistries have proven their ability to effect splice corrections of aberrant pre-mRNA transcripts [3] and restore protein expression in multiple clinical trials, and to date, five splice modifying ASO have received FDA approval to treat DMD, SMA, and Batten disease [3,4,5]. Splice modulation through exon skipping is currently the leading therapeutic strategy for the treatment of Duchenne Muscular Dystrophy (DMD) with two ASO chemistries evaluated in clinical trials for the targeting of DMD exon 45, 51, and 53: 2′-*O*-methyl-phosphorothioate (2′-OMe PS) and phosphorodiamidate morpholino oligomers (PMO) [6,7]. However, the therapeutic effect of clinical treatments stays relatively low as certain issues of ASO such as poor cellular up-take, fast clearance in the bloodstream, and toxicity remain to be solved [8,9]. A great variety of approaches have been studied during the years to enhance delivery and increase nuclease resistance of this new class of drugs [9,10]. We previously reported on the potential of certain cationic polymethylmethacrylate (PMMA) nanoparticles loaded with a 2′OMe PS ASO to induce dystrophin restoration in the muscles of mdx animal model [11]. Despite the positive results, slow biodegradability and tissue accumulation of PMMA nanoparticles limited their use in chronic treatments.

On the other hand, conjugation at the 3′-and/or 5′-ends with appropriate small molecules, which impart specific physical–chemical properties to ASO, represents an interesting strategy that could result in easier clinical development of ASO [10,12]. A vast array of molecules have been conjugated to increase exonucleases resistance, and to confer new properties, such as targeting capacity [13,14,15,16,17], increased lipophilicity [18,19,20], or membrane interaction [21]. Among the various functional molecules, hydrophobic natural compounds such as cholesterol, vitamin E, and fatty acids, represent an interesting choice, since it has been reported that they can improve ASO delivery in vitro by promoting endosomal release [22] and in vivo, not only to the liver, but also to peripheral muscle tissues [23]. Many synthetic strategies have been developed to link small functional molecules to oligonucleotides [24,25]; in particular, conjugations are most readily achieved at the 5′-end by adding an appropriate amino linker or a suitable phosphoramidite. Nevertheless, it is well-known that 3′-modified ASO usually show greater resistance to 3′-exonucleases, the major nucleases within cells and serum, compared to 5′-analogues. In addition, conjugation at 3′-end offers a different interaction with the splicing machinery in respect to that at the 5′-end. Several methods for the preparation of 3′-conjugated oligonucleotides have been reported [26,27,28,29,30]. Many of them related to post-synthetic approaches, where the cleavage from the solid support of newly assembled oligonucleotides generates a linker armed with a specific functional group able to bind the desired molecule [29]. Other methods involved the use of solid support armed with branched linkers allowing the introduction of desired molecules, both before and post-assembly of oligonucleotides [29,30]. However, there are just a few examples in the literature in which oligonucleotides have been conjugated with bile acids [28,31,32,33] and to the best of our knowledge, there are no reports of bile acid-conjugated ASO for splicing modulation approaches.

Here, we report the preparation of a 20-mer 2′-OMe PS oligonucleotide conjugated at 3′-end with ursodeoxycholic acid (ASO 51 3′-UDC, Figure 1), involved in the targeting of human DMD exon 51 [34]. We assumed that amphiphilic UDCA, a bioactive natural product, could improve the physical–chemical properties and then the activity of ASO 51 [35,36] which contains the sequence of the clinically evaluated Drisapersen oligonucleotide [37,38,39]. Regardless of the conventional role as a bio-surfactant, UDCA is also a long-established drug and exhibits anti-apoptotic properties towards different cell types [40]. Moreover, it has been shown that bile acids can improve bioavailability of highly hydrophilic drug molecules, by increasing the fluidity of biological membranes and promoting the chemical and enzymatic stability of drugs [35].

For the preparation of the desired ASO 51 3′-UDC, we designed two possible approaches. We started with a post-synthetic conjugation (Figure 2A). It consists of a solution phase reaction between activated-UDCA and ASO 51 armed at the 3′-end with an aminohexyl linker synthesized by using a commercially available support. The solution phase conjugation offers two major advantages: (i) the intermediate ASO (ASO 51 3′-C6-NH_2_) can be synthesized with standard protocols and purified in advance; (ii) the conjugation reaction is usually selective and scalable. However, the post-synthetic approach is quite lengthy and expensive, requiring an additional step of purification after the conjugation reaction of UDCA and the use of a costly commercial support. Envisaging a possible use of ASO 51 3′-UDC oligonucleotide for in vivo exon skipping studies, we designed as an alternative approach the pre-synthetic one (Figure 2B). It involves the synthesis of a novel solid support properly functionalized with UDCA before assembling the oligonucleotide, named UDC-S. Even though the process requires an appropriate derivatization of UDCA, it is synthetically simple, involves the use of safe and inexpensive commercially available materials, and avoids a solution phase conjugation after oligonucleotide synthesis.

We describe herein the synthesis of ASO 51 3′-UDC with the two synthetic approaches shown in Figure 2. The obtained UDC-conjugated ASO was evaluated for its hybridization property, cell toxicity, and exon-skipping efficiency.

## 2. Results

### 2.1. Synthesis of ASO 51 3′-UDC via Post-Synthetic Approach

For the synthesis of ASO 51 3′-UDC via the post-synthetic approach, the commercially available Primer Support^TM^ C6 Amino Linker 200 with a loading of 80 μmol/g was employed (Figure 2A). The oligonucleotide sequence was assembled following our previously reported standard protocol for the automated solid-phase synthesis of 2′OMe PS oligonucleotides via the phosphoramidite chemistry at 10 μmol scale (10). The sequence of ASO 51 targeting human DMD exon 51 was obtained from the literature [39]. Firstly, ASO 51 3′-C6-NH_2_ was synthesized in DMT-on mode and purified by ion pair HPLC (IP-HPLC), then after the removal of the DMT-group, dimethoxy-tritanol was decanted to obtain the desired ASO 51 3′-C6-NH_2_ oligonucleotide in 42% yield with the purity level required for the next reaction of coupling. To conjugate UDCA with ASO 51 3′-C6NH_2_ via amide linkage, a considerable optimization study was required. At first, we chose to carry out the conjugation by adding a coupling reagent such as 2-(1*H*-benzotriazol-1-yl)-1,1,3,3-tetramethyluronium hexafluorophosphate (HBTU) to 1.1 mM ASO 51 3′-C6-NH_2_ in DMF and UDCA exceeding (Figure 1A). The reaction proceeded speedily in the presence of diisopropylethyl amine (DIPEA), affording after 20 min the desired ASO 51 3′-UDC conjugate with a yield of 53% estimated by UV detection of ion pair HPLC (IP-HPLC) analysis of a sample. Nevertheless, due to ESI-MS analysis of a purified ASO 51 3′-UDC sample showing the presence of several unidentified byproducts, we evaluated the direct formation of an active ester as an alternative approach. Therefore, we investigated the conversion of UDCA into *N*-hydroxysuccinimide ester **1**. The reaction took place in 95% yield after crystallization of the crude compound by EtOAc. No further purification of **1** was carried out before conjugation in solution phase. A few attempts were made before obtaining the highest coupling yields in satisfactory reaction conditions (Appendix A). As shown in Figure 1B, by using a 5 mM solution of ASO 51 3′-C6-NH_2_ in DMSO, it was possible to employ only 2 equivalents of UDC-NHS **1** and 10 equivalents of DIPEA to achieve conjugated oligonucleotide with an 80% yield (by UV detection based on area of conjugated relative to unconjugated). Finally, ASO 51 3′-UDC was diluted in triethylammonium acetate buffer and first purified by IP-HPLC and then by ion exchange HPLC (IEX-HPLC) for sodium exchange to give the pure product in 30.6% overall yield. The molecular mass was verified by ESI-MS.

### 2.2. ASO 51 3′-UDC Is Stable in Cleavage/Deprotection Conditions

Before proceeding with pre-synthetic approach, the stability of amide group linking UDCA at the 3′-end of ASO 51 was tested in basic conditions required for deprotection/cleavage process (conc. aq. 33% NH_3_, 50 °C, 20 h). An aliquot of ASO 51 3′-UDC (10^−5^ M) was dissolved in 2 mL of conc. aq. NH_3_ and incubated at 50 °C for 20 h then analyzed by ion-pair HPLC and mass spectrometry. ASO 51 conjugated at 3′-end with UDCA resulted stable at this deprotection/cleavage condition, since no appreciable degradation was found in the examined sample (Appendix A).

### 2.3. Synthesis of ASO 51 3′-UDC via Pre-Synthetic Approach

#### 2.3.1. Ursodeoxycholic Acid Derivatization

For the synthesis of ASO 51 3′-UDC via pre-synthetic approach, we started with the functionalization of UDCA. The conversion of carboxylic group into *N*-hydroxysuccinimide ester **1** took place in 95% yield after crystallization and enabled the addition of 6-amino-1-hexanol linker through amidation affording UDC-hexanolamide 3 (Figure 2). Then, the primary hydroxy group of **3** was protected with DMT-Cl and the two secondary alcohols of DMT-compound **4** were subjected to a protection-deprotection strategy to allow the selective esterification with succinic anhydride of C3-OH. Therefore, the reaction with acetic anhydride in basic conditions provided diester **5** which, upon ethanolic KOH (0.089 M) hydrolysis of the most reactive group (the one in position 3), gave rise to monoester **6**. The reaction was monitored by TLC and was stopped after 3 h, even if the conversion of **5** into **6** was uncomplete. Prolonged times resulted in the formation of byproducts. Isolation of compound **6** was readily accomplished by flash chromatography with a 35% yield after three steps. Esterification of **6** into the hemisuccinate **2** was carried out in pyridine with 8 equivalents of succinic anhydride by heating at 70 °C. Compound **2** was now ready to be attached to the solid support. It is worth noting that, although the derivatization of UDCA involves six steps, only a fast purification by flash chromatography of crude monoester **6** was required to provide hemisuccinate **2** in good overall yield (24.7%) and purity level.

#### 2.3.2. Solid Support Functionalization: Synthesis of UDC-S Support

The commercial Primer Support^TM^ 5G Amino (GE Healthcare) with 350 µmol/g loading was deprotected following the producer’s recommendation. The resulting support with free amino groups was then functionalized with the modified bile acid **2**. All reactions carried out by using the coupling agent HBTU resulted in the formation of the desired UDC-S support only in very small traces (Table 1, entries 1,2). Activation of the hemisuccinate **2** as a pentafluorophenyl (PFP) ester also failed to give the desired functionalized support (Table 1, entry 3). Only by changing of the coupling agent, it was possible to achieve a satisfactory level of functionalization (Table 1, entry 4 and Figure 3). Additional experiments to improve the loading efficiency by increasing the equivalents of DIPEA (Table 1, entry 5) or of *O*-(1H-6-Chlorobenzotriazole-1-yl)-1,1,3,3-tetramethyluronium hexafluorophosphate (HCTU) (Table 1, entry 6) failed. Finally, unreacted amino groups of UDC-S support were acetylated by using commercially available capping solutions containing acetic anhydride (Figure 3). The loading efficiency of various reactions shown in Table 1 was evaluated by UV assay of DMT cation. The highest loading (Table 1, entry 4) was of 245 µmol/g, corresponding to a 70% of derivatization of starting support.

#### 2.3.3. Synthesis of ASO 51 3′UDC on UDC-S Support

Before carrying out the synthesis of the oligonucleotide of our interest, a coupling test with the phosphoramidite U was made to assess the synthesis conditions of a 2′OMe PS ASO on this new support, including the oxidation reaction to phosphorothioate. The test was performed by assembling a short tri-U following our standard protocol on a 3 µmol scale. The three coupling reactions proceeded in average yield of 100% per cycle as judged from the online software check of the DMT removal (by UV-Vis reading). Thus, we have successfully loaded the commercial polystyrene support with the desired functional molecule, then proceeded with a 50 µmol scale solid phase synthesis of ASO 51 3′-UDC (Figure 3). To this purpose, 205 mg of UDC-S support (245 µmol/g loading) were weighted in a 1.2 mL steel column and the synthesis of the target oligonuleotide was carried out in DMT-on mode. After reaching completion, the crude oligonucleotide was treated with ammonium hydroxide 33% at 50 °C for 24 h to allow the cleavage from the solid support and the removal of protecting groups. Analysis by ion-pair HPLC showed an overall UV-yield of crude DMT-on ASO 51 3′-UDC of 77.8%, which is a very good yield for a 3′-conjugated 2′OMe PS ASO. The crude DMT-on product was processed in standard conditions to give after HPLC purifications and ion exchange HPLC (IEX-HPLC) for sodium exchange pure ASO 51 3′-UDC in 45% total yield. The molecular mass was verified by ESI-MS (Appendix A).

### 2.4. ASO 51 3′UDC Has Improved Efficiency in Inducing Exon-Skipping without Toxicity

ASO 51 and ASO 51 3′-UDC were tested in myotubes derived from immortalized myoblasts bearing a deletion of the exon 52 in the dystrophin gene. The 28 cycles RT-PCR performed with primers on exon 50 and 54 of the dystrophin gene amplified in both conditions two products of 514 and 281 bp corresponding to the dystrophin transcripts lacking the exon 52 and both exons 51 and 52, respectively (Figure 3A,B). Exon skipping efficiencies resulted of ~5% in cells treated with ASO 51 and of 46% for the cells treated with the ASO 51 3′-UDC (Figure 3C). Cell toxicity evaluated by Trypan blue coloration determined the higher toxicity in cells treated with JetPEI alone (about 6% of dead cells) whereas cells treated with the two ASO had a similar number of dead cells (1.5 and 1.3%, respectively).

### 2.5. 3′-UDC Conjugation Is Not a Source of Duplex Instability

The melting temperatures (T_m_) of the duplex formed by and ASO 51 3′-UDC with the natural RNA complementary sequence was measured and compared with the corresponding duplex formed by unconjugated ASO 51. The ASO 51 RNA hybrid measured a T_m_ of 75.0 ± 0.8 and the ASO 51 3′-UDC-RNA duplex measured a T_m_ of 75.3 ± 0.3. These values suggested that UDCA conjugation did not destabilize the duplex (Appendix A).

### 2.6. ASO 51 and ASO 51 3′-UDC Aggregation Studies by Dynamic Light Scattering

DLS analysis of ASO 51 and ASO 51 3′-UDC in PBS showed very poor light scattering intensity with scattering correlograms of isolated oligomers having size smaller than 10 nm. No appreciable difference was noticed between the two investigated samples thus indicating that, at least in PBS, neither ASO 51 nor ASO 51 3′-UDC form micelles or nanoparticles.

## 3. Discussion

In the frame of our research work aimed at evaluating modified ASO for exon-skipping approaches [11,41], we decided to prepare a 2′OMe PS conjugated at 3′-end with UDCA targeting exon 51 in human DMD. Thanks to the well-established amphiphilic features of BAs, we envisaged of improving physical–chemical and pharmacokinetic properties of ASO 51 by its conjugation to UDCA [35,42,43]. We have targeted UDCA owing to certain feature, such as its anti-apoptotic property towards different cell types [36] and its long-established safety profile. Two different approaches have been designed (Figure 2A,B). In order to gain a quick insight about the synthesis of ASO 51 3′-UDC, we started with the most common post-synthetic approach. As final step, it envisaged the solution phase synthesis of an amide linkage between UDCA and ASO 51 armed at 3′-end with an aminohexyl linker (Figure 2A). To carry out the conjugation an optimization study was required, with the best result obtained by employing UDCA activated as NHS-ester **1**. As shown in Figure 1B, ASO 51 3′-UDC was obtained with an estimated coupling yield of 80% (based on UV area) when the reaction was carried out in DMSO and 2 equivalents of UDC-NHS **1**, in presence of 10 equivalents of DIPEA. Notably, the post-synthetic approach required no special synthetic efforts; nevertheless, it was quite lengthy and expensive. Indeed, ASO 51 3′-UDC synthesis needed the use of a costly modified support, and two cycles of purifications, i.e., one before and one after the coupling reaction with UDCA. However, ASO 51 3′-UDC was obtained in 30.6% overall yield and our first goal was achieved.

To evaluate an alternative approach most cost-effective for the synthesis of ASO 51 3′-UDC in view of a possible use of the oligonucleotide for in vivo exon skipping studies, we considered the pre-synthetic approach shown in Figure 2B. It is based on the functionalization of a polystyrene support achieved by amidation of the carboxylic group of UDCA with primary amino groups on the support. Then, before starting, we decided to test the stability of our ASO 51 3′-UDC including the amide linkage, under cleavage/deprotection conditions [44]. Fortunately, no valuable decomposition was detected by IP-HPLC and mass spectrometry analyses. Therefore, thanks to our well-established experience in the design and synthesis of multi-functional bile acid scaffolds [45,46], we focused on the derivatization of UDCA to prepare the ad-hoc modified polystyrene support UDC-S (Figure 2 and Figure 3). The addition of 6-amino-1-hexanol to UDC-NHS **1** allowed us to introduce the same C6-amino spacer present in ASO 51 3′-UDC obtained via post-synthetic approach. The following protection of primary alcohol as dimethoxytrityl ether was necessary to avoid its reaction during further functionalization of the bile acid and to have a reliable way of measuring the loading of the UDCA-modified solid support once obtained. The regioselective protection of the primary alcohol was ensured thanks to the hindered nature of the protective group chosen. It is worth noting that, although the derivatization of UDCA into hemisuccinate **2** involved six steps, only a simple purification by flash chromatography of crude monoester **6** was required to provide **2** with an overall yield of 24.7%. Hydrolysis of diester **5** into monoester **6** was the critical step dropping the yield of overall derivatization process. An optimization of this step failed as the basic conditions required for conversion of **5** into **6** resulted in the formation of several unidentified byproducts. A considerable study was required for the introduction of UDCA derivative **2** on the commercial Primer Support^TM^ 5G Amino. The highest loading yield providing UDC-S support with a loading of 245 µmol/g (Table 1, entry 4 and Figure 3) was obtained by using HCTU as a coupling reagent. On the other and, very few traces of UDC-S were obtained when the analogue HBTU aminium salt was employed or UDCA was activated as a PFP-ester. Then, we successfully tested the ability of UDC-S support to assemble a short tri-U following our standard protocol on a 3 µmol scale. Finally, a synthesis of ASO 51 3′-UDC on a 50 μmol scale was performed by using the newly prepared support. Notably, IP-HPLC analysis of crude DMT-on ASO 51 3′-UDC showed an UV-yield of 77.8% of the target compound, which is a very good yield for a 3′-conjugated 2′OMe PS ASO in solid phase. After IP-HPLC purification and sodium exchange, ASO 51 3′-UDC oligonucleotide was achieved with an overall yield of 45%; that confirms a more important output in comparison to that of post-synthetic approach.

To determine the ability of ASO 51 functionalized at 3′-end with UDCA to form a stable Watson–Crick duplex with its complementary RNA, UV melting curve analysis (260 nm) was carried out. From the experiment it appeared that 3′-UDCA conjugation did not destabilize the duplex, therefore, it could beneficially be exploited for antisense approaches.

Envisaging a potential use of ASO 51 3′-UDC for in vivo experiments, we investigated the ability of our amphiphilic ASO to spontaneously self-assemble into supramolecular systems. DLS analysis showed that ASO 51 3′-UDC did not form micelles or nanoparticles in physiological conditions, suggesting that the conjugation of a unit of UDCA is not enough to balance the hydrophilic nature of our 2′OMe PS ASO and facilitate the formation of supramolecular systems [47,48].

To functionally evaluate our product, the exon skipping efficiency of the conjugated oligonucleotide was compared with the one of ASO 51 by transfection in myogenic cell lines determining a better performance for the ASO 51 3′-UDC with an average 9.5-fold increase in respect to ASO 51. This suggests that, despite the inability to form micelles or nanoparticles, once entered in the cells, the ASO 51 3′-UDC has features enabling an increased ability to induce the skipping of the targeted exon compared to the unconjugated ASO. This might be related to either steric hindrance and/or an increased affinity with splicing negative regulators (e.g., hnRNPs, heterogeneous ribonucleoproteins) and has to be elucidated by further studies.

Low cytotoxicity is another beneficial feature of such conjugated oligonucleotide that will allow to vary the concentration of the ASO in a wide range and to optimize its therapeutic dosage for in vitro and in vivo studies.

## 4. Materials and Methods

Solvents were dried over molecular sieves EZ DRY moisture trap (EMP biotech), reactions were monitored by thin layer chromatography with silica gel 60 F245 (Merck), and spots were visualized with the aid of a phosphomolybdic acid solution. Flash chromatography was executed under nitrogen pressure with silica flash 230–400 mesh (Sigma Aldrich, Milano, Italy).

Solid phase oligonucleotide syntheses were carried out with fully automated oligosynthesizers (ÄKTA oligopilot 10 PLUS from GE Healthcare) controlled via UNICORN software. Commercially available solid supports: Primer Support^TM^ C6 Amino Linker 200, Primer Support^TM^ 5G Amino, and Primer Support^TM^ 5G Unylinker 350, were acquired from GE Healthcare. Batch reaction on solid supports were executed with the aid of a thermostatic oscillating stirrer Asal DVRL 711/CT. HPLC purifications were executed on ÄKTA purifier with reverse phase column Resource RPC 3 mL and HiScale 26/20 column packed with SOURCE 15RPC matrix. Ionic exchange was executed on HiTrap Capto S column. Lyophilization was carried out with Labconco FreeZone 1 connected to the centrifuge Christ RVC 2–18 CDplus.

^1^H-NMR (400 MHz) e ^13^C-NMR (101 MHz) were registered in the stated solvent on Varian Mercury Plus 400. ESI-MS spectra were obtained on spectrometer Thermo Finnigan LCQ Duo Ion Trap, and spectrophotometric analysis was executed with a Varian CARY 100 Bio instrument in the stated solvent.

### 4.1. General Procedure for Solid Phase Synthesis

2′-OMe PS oligoribonucleotides ASO 51 3′-UDC, ASO 51 3′-C6-NH_2_, and ASO 51 were synthesized at 50, 10, and 2 µmole scale, respectively, in DMT-on mode. For the synthesis of ASO 51 3′-C6-NH_2_ and ASO 51, commercial solid supports were employed; ASO 51 3′-UDC was synthesized with our functionalized UDC-S support. All synthesized ASO were full-length 2′-OMe phosphorothioates and contained the following sequence: 5′-UCAAGGAAGAUGGCAUUUCU-3′.

Detritylation was executed with a commercially available DCA deblock solution (3% DCA in toluene, Proligo Reagents); sulfurization was done with 0.2 M Phenyacetyldisulfide (Biosynth Carbosynth) in pyridine: acetonitrile (ACN) (1:1); the capping reaction was performed following the standard manufacturer’s protocols, with a Cap solution composed of: Cap A solution: *N*-methyimidazole in dry ACN 20% (*v*/*v*) and Cap B solution: 1:1 mixture of solution B1 (Ac_2_O in dry ACN 40% *v*/*v*) and solution B2 (2,6-Lutidine in dry ACN 60% *v*/*v*) (Proligo Reagents).

2′-OMe phosphoramidites (DMT-2′-OMe-rU-phosphoramidite, DMT-2′-OMe-rA-(bz)-phosphoramidite, DMT-2′-OMe-rG-(ibu)-phosphoramidite, and DMT-2′-OMe-rC-(ac)-phosphoramidite, ChemGenes) were dissolved in ACN (0.1 M) and the coupling reaction was carried out for 10 min, with BTT (5-benzylthio-1*H*-tetrazole 0.3 M in ACN, Biosynth Carbosynth), as an activator. Before cleavage, DMT-on oligonucleotides were treated with a solution of 20% diethylamine (DEA) in ACN for 10 min.

Cleavage from the solid support and removal of nucleobase protecting groups were achieved reacting the derivatized solid support, in a stoppered vial using 33% NH_4_OH aqueous solution (10 mL/g of resin) at 55 °C for 20 h. The support was then filtered off under vacuum and washed with H_2_O/EtOH (1:1) solution. The crude oligonucleotide yield was quantified by UV absorbance at 260 nm.

### 4.2. General Procedure for Oligonucleotide Purification

Purification of DMT-on oligonucleotides was carried out by ion pair HPLC (IP-HPLC) using ion pairing buffer conditions: buffer A: pH = 8.0 triethylammonium acetate (TEAA) with 5% ACN; buffer B: ACN, working in acetonitrile gradient (25% B in 6 column volumes, 45% B in 4 column volumes). DMT group removal was performed as follow: the lyophilized DMT-on oligonucleotide was dissolved in MilliQ water (1000 OD/mL) and left at room temperature overnight. The white precipitate of dimethoxytritanol was decanted after centrifugation. IP-HPLC analysis (ACN gradient: 15% in 2 column volumes, 45% in 4 column volumes) of the supernatant revealed complete removal of the protecting group. The lyophilized product was then re-hydrated and lyophilized again two times to strip TEAA residues. Finally, the oligonucleotide counterion (TEA^+^) was exchanged with sodium cation by ion exchange chromatography. The oligonucleotides, after purification steps, were quantified spectrophotometrically by UV absorbance at 260 nm.

### 4.3. Synthesis of ASO 51 3′-UDC by Coupling of UDC-NHS Ester ***1*** in DMSO

ASO 51 3′-C6-NH_2_ oligonucleotide was dissolved in DSMO to obtain a 5 mM solution and DIPEA (10 eq) and 2 eq of compound **1** (for its synthesis see Section 4.4 below) were added. The reaction was stirred at room temperature and monitored by IP-HPLC. After 18 h stirring, UV showed the formation of the conjugated oligonucleotide in 80% yield relative to unconjugated ASO 51 3′-C6-NH_2_. The reaction was diluted and purified by IP-HPLC (ACN gradient: 15% in 2 CV, 45% in 4 CV). The pure ASO 51 3′-UDC was obtained in 30.6% overall yield. MS (ESI, *m*/*z*): calculated for [C_241_H_327_N_77_O_124_P_20_S_20_] 7546.9; found 942.20 [M − 8H]^8−^, 1076.73 [M − 7H]^7−^, 1257.07 [M − 6H]^6−^, 1508.00 [M − 5H]^5−^.

### 4.4. Synthesis of UDC-NHS Ester ***1***

UDCA (10 g, 25.48 mmol, 1 eq) and *N*-hydroxy succinimide (3.5 g, 30.56 mmol, 1.2 eq) were dissolved in 200 mL anhydrous THF under nitrogen atmosphere. The solution was cooled to 0 °C in an ice bath and to this was added dropwise a solution of 6.28 g of *N,N’*-dicyclohexylcarbodiimide (DCC 30.56 mmol, 1.2 eq) dissolved in the minimum amount of anhydrous THF. The round bottom flask was gradually warmed until reaching 25 °C and the solution was stirred for 18 h at the same temperature. The reaction mixture was then filtered on Celite^®^, and the solvent removed under vacuum. The obtained solid was dissolved in EtOAc and the organic phase washed once with saturated NaHCO_3_ solution, once with distilled H_2_O and once with brine. The organic phase was dried by using anhydrous Na_2_SO_4_ and the filtered EtOAc evaporated under vacuum to give a white solid. The crude product was recrystallized from EtOAc to yield 12.02 g (24.58 mmol, 95%) of UDC-NHS ester **1** as an amorphous white solid.

MS (ESI, *m*/*z*) [M + Na]^+^: calculated for [C_28_H_43_NNaO_6_]^+^ 512.64; found 512.27.

^1^H-NMR (400 MHz; CDCl_3_) selected data: δ = 0.68 (d, 3H, *J* = 3.7 Hz, 18-CH_3_); 0.93 (s, 3H, 19-CH_3_); 0.94–0.98 (m, 3H, 21-CH_3_); 2.43–2.71 (m, 2H, 23-CH_2_); 2.83 (s, 4H, 2 CH_2_ of NHS); 3.51–3.65 (m, 2H, 3α-H, 7β-H).

^13^C-NMR (101 MHz; CDCl_3_) δ = 8.60; 12.09; 18.28; 21.14; 23.36; 25.38; 25.57; 26.84; 27.99; 28.55; 30.29; 30.61; 34.05; 34.89; 35.04; 36.79; 37.23; 39.12; 40.07; 42.39; 43.73; 45.77; 54.75; 55.65; 71.35; 71.44; 169.06; 169.20.

### 4.5. Synthesis of UDC-Aminohexyl Alcohol ***3***

An amount of 6.69 g of **1** (14.21 mmol, 1 eq) and 5.00 g of 6-amino-1-hexanol (42.65 mmol, 3 eq) were dissolved in 140 mL of dry DMF under nitrogen atmosphere. The solution was treated with 4.95 mL of DIPEA (28.44 mmol, 2 eq) and stirred for 24 h at 25 °C, then 140 mL of HCl 0.01 mM were added. The solution was extracted three times with DCM and the collected organic phases were dried with anhydrous Na_2_SO_4_, filtered, and evaporated under vacuum to achieve 6.53 g of **3** as an amorphous white solid with a yield of 93%.

MS (ESI, *m*/*z*) [M + H]^+^: calculated for [C_30_H_54_NO_4_]^+^ 492.77; found 492.60.

^1^H-NMR (400 MHz; CDCl_3_) selected data: δ = 0.66 (s, 3H, 18-CH_3_), 0.90–0.94 (m, 6H, 19-CH_3_, 21-CH_3_), 2.20 (ddt, 2H, *J* = 16.7; 11.2; 5.7 Hz, 23-CH_2_), 3.18–3.33 (m, 4H, 25-CH_2_, 30-CH_2_), 3.54–3.59 (m, 2H, 3α-H, 7β-H).

^13^C-NMR (101 MHz; CDCl_3_) δ = 12.11; 18.48; 21.15; 23.37; 25.26; 26.47; 26.88; 28.68; 29.65; 30.28; 31.85; 32.51; 33.64; 34.05; 34.90; 35.38; 36.81; 37.25; 39.15; 39.26; 40.13; 42.40; 43.73; 54.88; 55.71; 62.63; 71.34; 71.41; 173.57.

### 4.6. Synthesis of UDC-Amino-DMT-Hexanol ***4***

An amount of 3.27 g of **3** (6.65 mmol, 1 eq) was dried by dissolution and evaporation from anhydrous ACN twice. Then, the resulting solid was dissolved in 20 mL pyridine and 3.38 g of DMT-Cl (9.97 mmol, 1.5 eq) were added portionwise in 30 min. The mixture was stirred at 25 °C for 18 h and monitored by TLC (EtOAc) then, the mixture was used for the successive reaction in one pot.

An analytical sample of DMT-alcohol **4** was obtained after flash chromatography (EtOAc + 3% Et_3_N; Rf = 0.30).

MS (ESI, *m*/*z*) [M + Na]^+^: calculated for [C_51_H_71_NNaO_6_]^+^ 817.12; found 816.40.

^1^H-NMR (400 MHz; CD_3_OD) selected data: δ = 0.69 (s, 3H, 18-CH_3_); 0.94 (s, 3H, 19-CH_3_); 0.94–0.97 (m, 3H, 21-CH_3_); 2.99–3.16 (m, 4H, 25-CH_2_, 30-CH_2_); 3.42–3.51 (m, 2H, 3α-H, 7β-H); 3.77 (s, 6H, 2 OCH_3_); 6.80–6.85 (m, 4H, H in 3,3′,5,5′ of DMT); 7.14–7.20 (m, 1H, H in 4″ of DMT); 7.22–7.31 (m, 6H, H in 2,2′,2″,6,6′,6″ of DMT); 7.37–7.42 (m, 2H; H in 3″,5″ of DMT).

^13^C-NMR (101 MHz; CD_3_OD) δ = 11.28; 13.03; 17.61; 19.43; 20.96; 22.52; 25.79; 26.41; 26.53; 28.22; 28.95; 29.62; 32.00; 32.75; 33.73; 34.65; 35.31; 36.57; 36.18; 37.73; 38.87; 39.27; 40.13; 42.59; 43.06; 43.36; 54.24; 55.11; 56.06; 60.10; 62.85; 70.51; 70.68; 85.59; 106.41; 112.50; 126.18; 127.21; 127.86; 129.74; 136.43; 145.48; 147.70; 158.73; 175.21.

### 4.7. Synthesis of 3,7-Diacetyl-UDC-Amino-DMT-Hexanol ***5***

To the abovementioned reaction mixture were added 2.515 mL of acetic anhydride (26.6 mmol, 4 eq) and a catalytic amount of DMAP. The mixture was stirred at 25 °C for 18 h and monitored by TLC. The reaction was then quenched with the addition of 3 mL of CH_3_OH. After 30 min, the mixture was concentrated under vacuum and to the mixture were added water and DCM. The organic phase was washed with H_2_O, dried with anhydrous Na_2_SO_4_, filtered, and evaporated to give 3.44 g of crude compound **5** as a yellowish wax.

MS (ESI, *m*/*z*) [M + Na]^+^: calculated for [C_55_H_75_NNaO_8_]^+^ 901.19; found 900.47.

^1^H-NMR (400 MHz; CD_3_OD) selected data: δ = 0.58 (s, 3H, 18-CH_3_); 0.83–0.85 (m, 3H, 21-CH_3_); 0.86 (s, 3H, 19-CH_3_); 1.85 (s, 3H, CH_3_ Acetyl); 1.94 (s, 3H, CH_3_ Acetyl); 2.87–3.0 (m, 4H; 25-CH_2_, 30-CH_2_); 3.7 (s, 6H, 2 OCH_3_); 4.48–4.57 (m, 1H, 3α-H); 4.57–4.67 (m, 1H, 7β-H); 6.83–6.88 (m, 4H, H in 3,3′,5,5′ of DMT); 7.17–7.23 (m, 5H, of DMT); 7.24–7.37 (m, 4H, of DMT).

^13^C-NMR (101 MHz; CD_3_OD) δ = 11.71; 14.00; 18.28; 20.67; 20.96; 21.35; 22.69; 25.18; 25.51; 25.88; 26.17; 27.89; 29.02; 29.36; 31.50; 32.30; 32.42; 32.54; 33.43; 33.80; 34.55; 38.15; 38.48; 41.14; 42.96; 54.26; 54.40; 54.89; 62.60; 72.71; 72.86; 85.00; 87.25; 112.99; 126.43; 127.52; 127.66; 129.44; 135.95; 145.16; 157.83; 169.70; 169.75; 172.09.

### 4.8. Synthesis of 7-Acetyl-UDC-Amino-DMT-Hexanol ***6***

An amount of 3.44 g (3.92 mmol, 1 eq) of compound **5** was dissolved in 88 mL of KOH solution 0.089 M in EtOH (7.83 mmol, 2 eq). The solution was stirred at 25 °C for 3 h, and then a solution of phosphate buffer was added to reach a pH between 7–8, before extraction with EtOAc. The organic phase was then dried with anhydrous Na_2_SO_4_, filtered and the solvent evaporated under vacuum, to afford 3.28 g of crude **6** which was purified by flash chromatography (EtOAc/cyclohexane 3:1 + Et_3_N 3%. (Rf = 0,37) to give 1.29 g of pure compound **6** as a white waxy solid. The yield calculated over three steps was 35%.

MS (ESI, *m*/*z*) [M + Na]^+^: calculated for [C_53_H_73_NNaO_7_]^+^ 859.16; found 859.15.

^1^H-NMR (400 MHz; CD_3_OD) selected data: δ = 0.69 (s, 3H, 18-CH_3_); 0.92 (s, 3H, 19-CH_3_); 0.95 (s, 3H, 19-CH_3_); 1.92 (s, 3H, CH_3_ Acetyl); 2.99–3.18 (m, 4H, 25-CH_2_, 30-CH_2_); 3.41–3.56 (m, 1H, 3α-H); 3.77 (s, 6H, 2 OCH_3_); 4.74 (td, 1H, *J* = 11.2; 5.2 Hz, 7β-H); 6.80–6.86 (m, 4H, H in 3,3′,5,5′ DMT); 7.13–7.19 (m, 1H, H in 4″ DMT); 7.21–7.31 (m, 6H, H in 2,2′,2″,6,6′,6″ DMT); 7.36–7.43 (m, 2H, *J* = 8.5, 1.8 Hz, H in 3″,5″ DMT).

^13^C-NMR (101 MHz; CD_3_OD) δ = 12.58; 18.92; 21.78; 22.30; 22.83; 23.76; 26.92; 27.27; 27.86; 29.50; 30.39; 30.91; 31.08; 33.33; 34.10; 35.03; 35.87; 36.52; 37.76; 40.26; 40.72; 41.26; 43.63; 44.74; 55.67; 56.42; 56.70; 64.29; 71.92; 75.32; 87.06; 113.93; 127.61; 128.64; 129.28; 131.15; 137.85; 146.99; 159.94; 172.55; 176.55.

### 4.9. Synthesis of 3-Hemisuccinyl-7-Acetyl-UDC-Amino-DMT-Hexanol ***2***

Amounts of 1.29 g of compound **6** (1.54 mmol, 1 eq), 1.23 g of succinic anhydride (12.34 mmol, 8 eq), 0.75 g of DMAP (6.17 mmol, 4 eq) were dissolved in 20 mL of pyridine under nitrogen atmosphere. The reaction mixture was stirred for 18 h at 70 °C, after which, the solvent was evaporated under vacuum. The solid was redissolved in EtOAc, washed with saturated NaHCO_3_ solution, and then with citric acid 0.5% in water. The organic phase was dried by anhydrous Na_2_SO_4_, filtered and the solvent removed under vacuum, to give 1.14 g (80% yield) of desired compound **2** as an amorphous white solid which was employed in the next reaction without any further purification. An analytical sample was obtained by flash chromatography (EtOAc/cyclohexane 9:1).

MS (ESI, *m*/*z*) [M − H]^−^: calculated for [C_57_H_76_NO_10_]^−^ 935.23; found 934.67.

^1^H-NMR (400 MHz; CD_3_OD) selected data: δ = 0.69 (s, 3H, 18-CH_3_); 0.94 (s, 3H, 19-CH_3_); 0.96 (s, 3H, 21-CH_3_); 1.92 (s, 3H, CH_3_ Acetyl); 2.56 (m, 4H, 2 CH_2_ Succinyl); 3.01–3.2 (m, 4H, 25-CH_2_, 30-CH_2_); 3.77 (s, 6H, 2 OCH_3_); 4.6–4.69 (m, 1H, 3α-H); 4.74 (dd, 1H, *J* = 10.7; 6.0 Hz, 7β-H); 6.79–6.86 (m, 4H, H in 3,3′,5,5′ DMT); 7.07–7.3 (m, 7H, DMT); 7.41 (dd, 2H, *J* = 10.0; 8.6, DMT).

^13^C-NMR (101 MHz; CD_3_OD) δ = 12.55; 18.92; 21.76; 22.30; 23.63; 26.92; 27.29; 27.37; 27.89; 29.50; 29.78; 30.40; 31.10; 33.31; 33.87; 34.09; 35.07; 35.46; 36.52; 40.27; 40.63; 41.19; 43.41; 44.73; 55.67; 56.33; 56.57; 64.30; 75.12; 75.34; 87.02; 113.94; 127.62; 128.65; 129.29; 131.17; 137.85; 159.96; 172.55; 173.76; 176.57.

### 4.10. Synthesis of UDC-S Support

An amount of 770 mg of Primer Support^TM^ 5G Amino (loading: 350 ÷ 400 μmol/g; 0.272 mmol) was loaded in a steel column of 6.2 mL volume and a 2% DIPEA solution in ACN was fluxed at 1 mL/min for 30 min; then, the support was washed by fluxing ACN at 2 ml/min until the conductivity reached 0 µS and dried under nitrogen. In a round bottom flask, 280 mg (0.299 mmol, 1.1 eq) of anhydrous compound **2** were dissolved in 6 mL of anhydrous ACN and the activated support was added and the flask swirled for 5 min in a thermostatic oscillating stirrer, then 104 µL of anhydrous DIPEA (0.598 mmol, 2 eq) were added and the swirling continued for 15 min. A solution of HCTU (124 mg, 0.299 mmol, 1 eq) dissolved in 2 mL of anhydrous ACN was added and the reaction mixture swirled overnight at 25 °C. The support was filtered and washed two times with 5 mL of ACN and two times with the same amount of DCM, then dried under vacuum at 40 °C overnight. To a 250 mL round bottom flask containing 5 mL of Cap A solution and 5 mL of Cap B solution, the functionalized support was added and the flask swirled overnight at 25 °C, and the support washed with ACN, DCM, CH_3_OH and H_2_O in succession. After drying under vacuum at 40 °C for 18 h, 725 mg of the desired functionalized UDC-S support were obtained.

### 4.11. Loading Determination

The UDC loading was determined measuring the acidic release of the bounded DMT group. To 100 mg of UDC-S support a 0.1 M solution of *p*-toluenesulfonic acid in ACN was added in a flask until 500 mL of total volume. The suspension was placed in an ultrasonic bath for 10 min, then left to decant for 1 h and the supernatant analyzed by UV-Vis spectrophotometer at 411 nm wavelength. The estimated loading of UDC-S was 245 µmol/g.

### 4.12. Synthesis of ASO 51 3′-UDC on UDC-S Support

ASO 51 3′-UDC was synthesized by using UDC-S support at 50 µmole scale as described above in the general procedure. The oligonucleotide yield after purification steps, was 45%. MS (ESI, ES-): calculated for [C_241_H_327_N_77_O_124_P_20_S_20_] 7546.9;. found 1076.72 [M − 7H]^7−^, 1256.67 [M − 6H]^6−^, 1507.88 [M − 5H]^5−^.

### 4.13. ASO 51 3′-UDC Stability in Cleavage/Deprotection Conditions

The stability of ASO 51 3′-UDC was tested in deprotection/cleavage conditions (conc. aq. NH_3_, 50 °C, 20 h); IP-HPLC analysis were executed on ÄKTA purifier with reverse phase column Resource RPC 3 mL using ion pairing buffer conditions: buffer A: pH = 8 triethylammonium acetate (TEAA) with 5% ACN; buffer B: ACN, working in acetonitrile gradient (ACN gradient: 15% in 2 CV, 45% in 4 CV).

### 4.14. Melting Studies

UV melting curves were determined at 260 nm on a Varian Cary 100 spectrophotometer that was equipped with a Peltier block using the Varian WinUV software. Complementary RNAs were mixed to 1:1 stoichiometry with a 2 μM single strand oligonucleotide concentration (in 10 mM phosphate buffer NaH_2_PO_4_ (pH 7), 150 mM NaCl). The melting curves (absorbance versus temperature) were recorded at 260 nm by applying a heating–cooling–heating cycle in the temperature range of 25–90 °C at a sweep rate of 0.4 °C/min. All experiments were done in triplicates.

### 4.15. Dynamic Light Scattering (DLS) Characterization

Dynamic Light Scattering (DLS) analysis was accomplished using a NanoBrook Omni Particle Size Analyser (Brookhaven Instruments Corporation, Holtsville, NY, USA) equipped with a 35 mW red diode laser (nominal 640 nm wavelength) in backscattering mode. ASO 51 and ASO 51 3′UDC were solubilized in PBS 0.15 M and analyzed at 25 °C. Samples’ concentration was ~0.07–0.08% wt. (0.1 mM). Each measurement was repeated five times on the same sample.

### 4.16. Exon-Skipping and Toxicity Studies

Immortalized myoblasts obtained from a patient with exon 52 deletion (Δ52) and kindly provided by Professor Vincent Mouly were maintained in Skeletal Muscle Growth Medium (Promocell) and differentiated into myotubes for at least 7 days in Skeletal Muscle Differentiation Medium (Promocell) supplemented with 2% horse serum at 37 °C with 5% CO_2_ incubation. Myotubes were transfected in 24-well plates with either 2 µg of ASO 51 or ASO 51 3′UDC using 4 uL of JetPEI (Polyplus).

Cell toxicity was determined by inspecting the cells and Trypan blue vital count. Cells treated with only JetPEI represented the controls. 24 and 48 h after the treatment cells were harvested and stained 1:1 with 0.4% Trypan blue vital dye (Sigma-Aldrich, Milano, Italy). Viable and dead cells were counted in a Burker hemocytometer under a light microscope.

Total RNA was isolated from myotubes two days after ASO transfection using the RNeasy Kit (QIAGEN) and reverse-transcribed by means of a High-Capacity cDNA Reverse Transcription Kit (Applied Biosystems), according to the manufacturer’s instructions. Before cDNA synthesis, RNA was treated with DNase I (Roche) and assessed for residual DNA contamination by a 55-cycle PCR.

Twenty-eight cycles RT-PCRs were performed with primers designed on exon 50 (DMDex50F CCTGACCTAGCTCCTGGACT) and exon 54 (DMDex54R GTCTGCCACTGGCGGAGGTC) reported in the http://dmd.nl/ website (accessed on 4 November 2021) and checked on a 1% gel. One microliter of the RT-PCR was loaded on High-sensitivity DNA chips (Agilent) for the quantification of the exon skipping which was performed calculating the ratio of the area of the skipped transcript and the sum of the area of unskipped and skipped transcripts multiplied for 100. Experiments were performed in triplicates. Significance was evaluated by paired t-test with post hoc Mann–Whitney test for *p* values less than 0.005 considered significant.

## 5. Conclusions

In conclusion, the preparation of a 20-mer 2′-OMe PS oligonucleotide conjugated at 3′-end with UDCA exploitable in the targeting of human DMD exon 51 has been reported. Both a pre-synthetic and a solution phase approach were successfully applied for its synthesis. While the solution method allows for more modularity, the pre-synthetic method is straightforward ensuring better yields, due to fewer unit operations after solid phase synthesis than the solution-phase approach. Easy chemical modifications of the bile acid allowed the production of a stable modified solid support in gram quantities starting from a cheap amino functionalized polystyrene. The pre-synthetic approach requires only a chromatographic purification to yield a product suitable for biological experiments. By converse, post synthetic functionalization required two, time consuming, chromatographic purifications with a prolonged manipulation causing concerns of degradation and sterility on the modified ASO and the purchase of an expensive amino linker modified solid support. The scale up of this procedure would be relatively cost effective as bile acids and amino modified resins result more affordable than linker bearing solid supports does. Our pre-synthetic procedure could be adapted for other bile acid conjugations to oligonucleotides and/or for the solid phase synthesis of other oligonucleotides incorporating different chemistries. Other linkers besides 6-amino-hexanol could be employed in order to minimize the steric hindrance between UDCA and the oligonucleotide and improve the exon-skipping efficiency.

The ASO 51 3′-UDC was characterized by no toxicity in human immortalized myogenic cells, good ability to form a stable duplex with its complementary RNA, and a 9.5-fold increase in exon-skipping efficiency compared to the unconjugated ASO. While toxicology and mechanistic studies are still pending to confirm the safety profile and mode of action before the in vivo tests, these results highlights the properties of UDCA-3′-end conjugated ASO, making them particularly interesting for exon skipping-based therapeutic approaches.

## 6. Patents

The work reported in this article is protected by the patent WO 2020/084488 Al 2020.

## Data Availability

Data sharing is not applicable to this article.

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
