# Peer review of "Synthesis and Exon-Skipping Properties of a 3′-Ursodeoxycholic Acid-Conjugated Oligonucleotide Targeting *DMD* Pre-mRNA: Pre-Synthetic versus Post-Synthetic Approach"

_molecules, 2021, doi:10.3390/molecules26247662_

Round 1
Reviewer 1 Report
The manuscript describes the synthesis and exon-skipping properties of ASO51-3’-UDCA for the inhibition DMD exon 51. From the synthesis point of view, the goal is challenging.
Two different strategies have been tested to introduce this bioactive bile acid into therapeutic ODN with successful results. The pre-synthetic approach seems more efficient although post-synthetic method is valuable as in some cases could be the only method to conjugate specific molecules to ODN.
Important efforts of nucleic acids laboratories have been devoted to produce in large scale modified oligonucleotide to stabilize its structure, improve the inhibition properties as well as resistance to biodegradation. Conjugations to lipids moieties or amphiphilic molecules are especially interesting for improving the delivery of ODN without compromising the inhibitory properties. The advances carried out in this field contributes to the improvement of therapeutic oligonucleotides.
The paper is well described and I support publication. Two typos: hemisuccinate (line 183) , dicyclohexylcarbodiimide (line 406) and (O-(1H-6-Chlorobenzotriazole-1-yl)-1,1,3,3-tetramethyluronium hexafluorophosphate ) should be added to the abbreviation HCTU (line 188).
Author Response
Response to comments and suggestions:
We thank the Reviewer for giving us the opportunity to submit a revised draft of the manuscript “Synthesis and Exon-Skipping Properties of a 3’-Ursodeoxycholic Acid-Conjugated Oligonucleotide Targeting DMD pre-mRNA: a Pre-Synthetic Approach for Incorporation of UDCA” for publication in Molecules. We have appreciated and incorporated all suggestions made by the Reviewer. Please, see below, in blue, for a point-by-point response.
1) Line 183: hemisuccinate
Author response: we have corrected as suggested
2) Line 188: (O-(1H-6-Chlorobenzotriazole-1-yl)-1,1,3,3-tetramethyluronium hexafluorophosphate ) should be added to the abbreviation HCTU
Author response: we have corrected as suggested
3) Line 406: dicyclohexylcarbodiimide
Author response: we have corrected as suggested

Reviewer 2 Report
This is an interesting report on the preparation of 3'-end modified oligonucleotide(s) using modified solid support that simplifies as well as economizes such syntheses when compared with post-synthetic modification protocols more often used. At this same time, it is the first report on the use of title modification (ursodeoxycholic acid as amide) to attach to the 3'-end of the oligonucleotides and shown to improve their exon skipping efficiency. This might be of practical importance in experimental therapies.
There are minor errors in the manuscript that are listed below.
The first is the TITLE where "Ursodeoxcholic" should be replaced by "Ursodeoxycholic".
Line 183 - is the emisuccinate 2; should be: the hemisuccinate 2
The same error is in supplementary file - page 5
Lines 186-188 - Table 2 can not be found in the manuscript ... Then, where is this table? Perhaps the authors refer to Table 1.
Line 196 - This is NOT "UV assay" to determine DMT - this is vis range (DMT+ cation solution is orange), so UV-Vis is advised.
Line 209 - as above.
The above list might not be complete and thus, a careful text edition is strongly advised.
Author Response
Response to comments and suggestions:
We thank the Reviewer for giving us the opportunity to submit a revised draft of the manuscript “Synthesis and Exon-Skipping Properties of a 3’-Ursodeoxycholic Acid-Conjugated Oligonucleotide Targeting DMD pre-mRNA: a Pre-Synthetic Approach for Incorporation of UDCA” for publication in Molecules. We have appreciated and incorporated all suggestions made by the Reviewer. Please, see below, in blue, for a point-by-point response.
1) TITLE: where "Ursodeoxcholic" should be replaced by "Ursodeoxycholic".
Author response: we have corrected as suggested
2) Line 183: the emisuccinate 2 should be: the hemisuccinate 2
Author response: we have corrected as suggested
3) The same error is in supplementary file - page 5
Author response: we have corrected as suggested
4) Lines 186-188: Table 2 can not be found in the manuscript ... Then, where is this table? Perhaps the authors refer to Table 1
Author response: We thank the Reviewer for this comment. Table 2 is a typo. We replaced it with Table 1
5) Lines 196 and 209 - This is NOT "UV assay" to determine DMT - this is vis range (DMT+ cation solution is orange), so UV-Vis is advised
Author response: we have appreciated the Reviewer’s comment and corrected as advised.
6) The above list might not be complete and thus, a careful text edition is strongly advised
Author response: We thank the Reviewer for this suggestion. A careful checking text has been made

Reviewer 3 Report
In the manuscript entitled “Synthesis and Exon-Skipping Properties of a 3’-Ursodeoxcholic Acid-Conjugated Oligonucleotide Targeting DMD pre-mRNA: a Pre-Synthetic Approach for Incorporation of UDCA by Marchesi et al., the authors tested a new approach to produce an ASO based on UDCA for skipping mutated exon 51 of the dystrophin gene. The manuscript is well written, clear, and achieved the aims proposed. There are no major comments and just one minor comment. Regarding the probability of alpha level established to determine the statistical difference, have the Authors set on 0.05 (5%) or 0.005 (0.5%).
Author Response
Response to comments and suggestions:
We thank the Reviewer for giving us the opportunity to submit a revised draft of the manuscript “Synthesis and Exon-Skipping Properties of a 3’-Ursodeoxycholic Acid-Conjugated Oligonucleotide Targeting DMD pre-mRNA: a Pre-Synthetic Approach for Incorporation of UDCA” for publication in Molecules.
We thank Reviewer for his comment, below, in blue, our response.
The probability of Alpha level was set to a standard 5% (0.5). We wrote two asterisks because the calculated p value resulted lower than 0.005 (p=0.0047 as reported in the figure legend).
